# Investigating Italian disinformation spreading on Twitter in the context of 2019 European elections

**Francesco Pierri** *, **Alessandro Artoni, Stefano Ceri**

Dept. of Electronics, Information and Bioengineering, Politecnico di Milano, Milano, Italy

* francesco.pierri@polimi.it

## Abstract

We investigate the presence (and the influence) of disinformation spreading on online social networks in Italy, in the 5-month period preceding the 2019 European Parliament elections. To this aim we collected a large-scale dataset of tweets associated to thousands of news articles published on Italian disinformation websites. In the observation period, a few outlets accounted for most of the deceptive information circulating on Twitter, which focused on controversial and polarizing topics of debate such as immigration, national safety and (Italian) nationalism. We found evidence of connections between Italian disinformation sources and different disinformation outlets across Europe, U.S. and Russia, featuring similar, even translated, articles in the period before the elections. Overall, the spread of disinformation on Twitter was confined in a limited community, strongly (and explicitly) related to the Italian conservative and far-right political environment, who had a limited impact on online discussions on the up-coming elections.

## Introduction

In recent times, growing concern has risen over the presence and the influence of deceptive information spreading on social media [1]. The research community has employed a variety of different terms to indicate the same issue, namely disinformation, misinformation, propaganda, junk news and false (or "fake") news.

As peopleare more and more suspicious towards traditional media coverage [2], news consumption has considerably shifted towards online social media; these exhibit unique characteristics which favored, among other things, the proliferation of low-credibility content and malicious information [1, 2]. Consequently, it has been questioned in many circumstances whether and to what extent disinformation news circulating on social platforms impacted on the outcomes of political votes [2–5].

Focusing on 2016 US Presidential elections, recent research has shown that false news spread deeper, faster and broader than reliable news [6], with social bots and echo chambers playing an important role in the diffusion of deceptive information [7, 8]. However, it has also been highlighted that disinformation only amounted to a negligible fraction of online news

**Data Availability Statement:** Data is publicly accessible on Dataverse at: https://doi.org/10.7910/DVN/OQHLAJ.

**Funding:** F.P. and S.C. are supported by the PRIN grant HOPE (FP6, Italian Ministry of Education). S.

C. is partially supported by ERC Advanced Grant 693174. The funders had no role in study design, data collection and analysis, decision to publish, or preparation of the manuscript.

**Competing interests:** The authors have declared that no competing interests exist.

[9–11], the majority of which were exposed to and shared by a restricted community of old and conservative leaning people, highly engaged with political news [9–11]. In spite of such small volumes, a study suggested that false news (and the alleged interference of Russian trolls) played an important role in the election of Donald Trump [2].

As the European Union (EU) struggled to counter the financial crisis which took place since the end of 2009 (following 2008 financial crisis in the US), populist and anti-establishment movements emerged as new electoral forces in Europe [12]. After the 2016 Brexit Referendum, anti-Europeans parties spread across the continent defining national identities in terms of ethnicity and religion and supporting tighter immigration controls [13]. As Europeans were called to elect their new representatives at the European Parliament–between the $23^{rd}$ and the $26^{th}$ of May 2019–populist and nationalist parties contrasted more traditional ones, such as European People's Party (EPP), Socialists and Democrats (S&D) and Alliance of Liberals and Democrats for Europe (ALDE), generally engaged in the defense of fundamental values associated with the EU project. Eventually, the pro-European side prevailed on aforementioned disruptive forces in most countries, but not in Italy where "Lega" amplified its electoral consensus (33%) and instead "Movimento 5 Stelle" declined (18%). Outside of our scope, a change of the Italian government occurred during the Summer of 2019.

For what concerns misbehavior on social platforms in European countries, recent research has highlighted the impact and the influence of social bots and online disinformation in different circumstances, including 2016 Brexit [5], 2017 French Presidential Elections [4, 14] and 2017 Catalan referendum [15]. A significant presence of disinformation in online conversations concerning 2019 European elections has been recently reported across several countries [14, 16–18]. The European Commission has itself raised concerns–since 2015 [19]–about the large exposure of citizens to disinformation, promoting an action plan to build capabilities and enforce cooperation between different member states. In anticipation of 2019 European Parliament elections, they sponsored an ad-hoc fact-checking portal (www.factcheck.eu) to debunk false claims relative to political topics, aggregating reports from several agencies across different countries.

For what concerns Italy, according to Reuters [20], trust in news is today particularly low (40% of people trust overall news most of the time, 23% trust news in social media most of the time), as result of a long-standing trend which is mainly due to the political polarization of mainstream news organizations and of the resulting partisan nature of Italian journalism. Previous research on online news consumption highlighted the existence of segregated communities [21] and explored the characteristics of polarizing and controversial topics which are traditionally prone to misinformation [22]. Remarkable exposure to online disinformation was highlighted by authors of [23], who exhaustively investigated online media coverage in the run-up to 2018 Italian General elections; in particular, the study observed a rising trend in the spread of malicious information, with a peak of interactions in correspondence with the Italian elections. This result was later substantiated in a report of the Italian Authority for Communications Guarantees (AGCOM) [24]. A very recent work [25] has collected electoral and socio-demographic data, relative to Trentino and South Tyrol regions, as to directly estimate the impact of false news on the 2018 electoral outcomes, with a focus on the populist vote; this study argues that malicious information had a negligible and non-significant effect on the vote. Furthermore, a recent investigation by Avaaz [26] revealed the existence of a network of Facebook pages and fake accounts which spread low-credibility and inflammatory content–reaching over a million interactions–in explicit support of "Lega", "Movimento 5 Stelle" about controversial themes such as immigration, national safety and anti-establishment. Those pages were eventually shut down by Facebook as violating the platform's terms of use.

In this work we focus on the 5-month period preceding 2019 European elections; wecarry out our research on a consolidated setting, described in [8, 27, 28], for investigating the presence (and the impact) of disinformation in the Italian Twittersphere. We recognize that our analysis has a few inherent limitations: first, according to Reuters [20] Twitter is overtaken by far by other social platforms, accounting for only 8% of total users (with a decreasing trend) when it comes to consume news online compared to Instagram (13%), YouTube (25%), WhatsApp (27%) and Facebook (54%), which exhibit instead a rising trend. Second, these differences are even more accentuated when comparing with the U.S. scenario [24], the focus of most of recent research. However, other aforementioned social media offer today little opportunities to researchers to conveniently analyze the spread of online information, given the limitations they impose on the acquisition of data and the different user experiences they offer. Our study sheds light on the Italian mechanisms of disinformation spreading, and thus the outcomes of the analysis indicate directions for future research in the field.

To collect relevant data, we manually curated a list of websites which have been flagged by fact-checking agencies for fabricating and spreading a variety of malicious information, namely inaccurate and misleading news reports, hyper-partisan and propaganda stories, hoaxes and conspiracy theories. Differently from [8], satire was excluded from the analysis. Following literature on the subject [3, 7, 9–11], we used a "source-based" approach, and assumed that all articles published on aforementioned outlets indeed carried deceptive information; nonetheless, we are aware that this might not be always true and reported cases of misinformation on mainstream outlets are not rare [3]. Our analysis was driven by the following research questions:

**RQ1**. What was the reach of disinformation which circulated on Twitter in the run-up to European Parliament elections? How active and strong is the community of users sharing disinformation?

**RQ2**. What were the most debatedthemes of disinformation? How much were they influenced by national vs European-scale topics?

**RQ3**. Who are the most influential spreaders of disinformation? Do they exhibit precise political affiliations? How could we dismantle the disinformation network?

**RQ4**. Do disinformation outlets share deceptive content in a coordinated manner? Can we identify connections with websites from other countries?

We first describe the data collection and the methodology employed to perform our analysis, then we discuss each of the aforementioned research questions, and finally we summarize our findings.

## Methods

### Data collection

Following a consolidated strategy [7, 8, 27, 28], we leveraged Twitter Streaming API in order to collect tweets containing an explicit Uniform Resource Locator (URL) associated to news articles shared on a set of Italian disinformation websites. As a matter of fact, using the standard streaming endpoint allows to gather 100% of shared tweets matching the defined query [8, 27, 28].

To this aim we manually compiled a list of 63 disinformation websites that were still active in January 2019. We relied on blacklists curated by local fact-checking organizations (such as "butac.net", "bufale.net" and "pagellapolitica.it"); these include websites and blogs which share hyper-partisan and conspiratorial news, hoaxes, pseudo-science and satire. We initially started

with only a dozen of websites, and we successively added other sources; this did not alter the overall collection procedure.

For sake of comparison, we also included four Italian fact-checking and debunking agencies, namely "lavoce.info", "pagellapolitica.it", "butac.net", "bufale.org".

In accordance with current literature [6, 9–11, 27, 28] we use a "source-based" approach: we do not verify each news article manually but we assign the *disinformation* label to all items published on websites labeled as such (the same holds for *fact-checking* articles).

In order to filter relevant tweets, we used all domains as query `filter` parameters (dropping "www", "https", etc) in the form "`byoblu com OR voxnews info OR ...`" as suggested by Twitter Developers guide (https://developer.twitter.com). We built a crawler to visit these websites and parse URLs as to extract article text and other metadata (published date, author, hyperlinks, etc). We handled URL duplicates by directly visiting hyperlinks and comparing the associated HTML content. We also extracted profile information and Twitter timelines for all users using Twitter API.

The collection of tweets containing disinformation (see Fig 1) and fact-checking articles was carried out continuously from January 1st (2019) to May 27th, the day after EU elections in Italy. We collected 16,867 disinformation articles shared over 354,746 tweets by 23,243 unique users, and 1,743 fact-checking posts shared over 23,215 tweets by 9814 unique users.

We can observe that, in general, articles devoted to debunk false claims were barely engaged, accounting only for 6% of the total volume of tweets spreading disinformation in the same period; such findings are comparable with the US scenario [8], and they are in accordance with the very low effectiveness of debunking strategies which is documented in [29]. We leave for future research an in-depth comparative analysis of diffusion networks pertaining to the two news domains.

The entire data is available at: https://doi.org/10.7910/DVN/OQHLAJ.

**Comparison with Facebook.** In order to perform a rough estimate of the different reach of disinformation on Twitter compared to Facebook, we collected data relative to the latter platform regarding two disinformation outlets, namely "byoblu.com" and "silenziefalsita.it", which have an associated Facebook page and are among Top-3 prolific and engaged sources of malicious information (see Results).

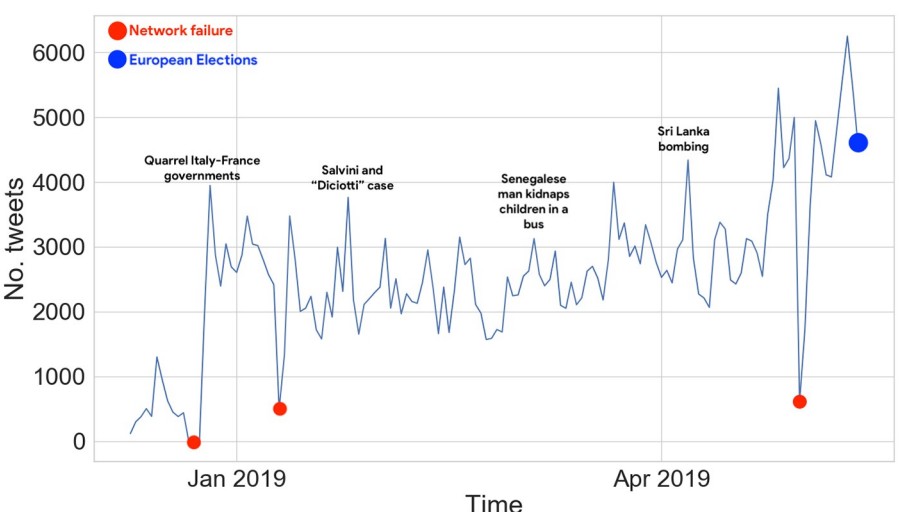

**Fig 1. Time series for the number of tweets, containing links to disinformation articles, collected in the period from 07/01/2019 to 27/05/2019.** We annotated it with some events of interest; network failures indicate when the collection tool went down.

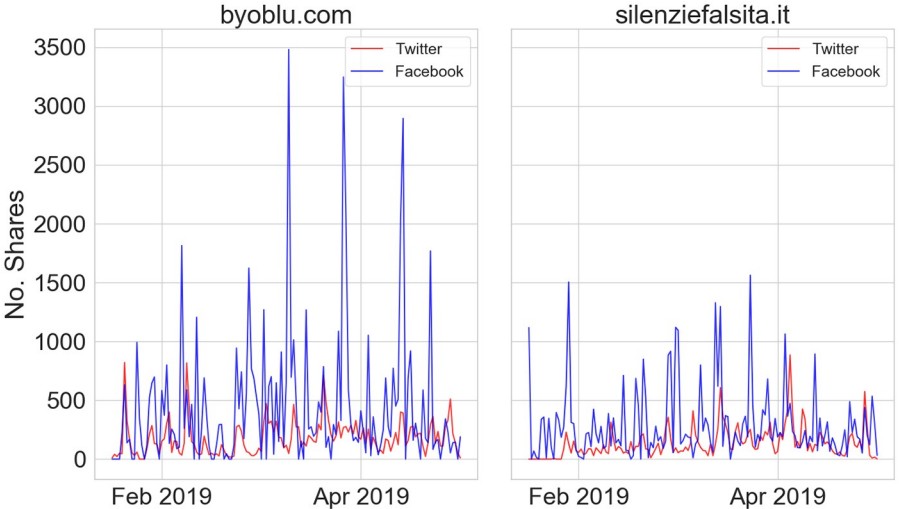

**Fig 2. Time series for the number of shares on both Twitter (red) and Facebook (blue) for two disinformation outlets, respectively "byoblu.com" (left) and "silenziefalsita.it" (right), in the period from 07/01/2019 to 27/05/2019.**

We used `netvizz` [30] to collect statistics on the number of daily shares of Facebook posts published by aforementioned outlets, and we compared with the traffic observed on Twitter. As we can see in Fig 2, disinformation has a stronger reach on Facebook than Twitter, for both sources, throughout the observation period; this is alsoshown in other works [23, 24, 26], coherently with the Italian consumption of social news. An in-depth analysis of the Italian disinformation on Facebook would be required, but it needs suitable assistance from Facebook for what concerns the disinformation diffusion network.

## Network analysis

**Building Twitter diffusion network.** We builtTwitter global diffusion network–corresponding to the union of all sharing cascades associated to articles gathered in our dataset–following a consolidated strategy [7, 8]. We considered different Twitter social interactions altogether and for each tweet we add nodes and edges differently according to the action(s) performed by users:

- **Tweet**: a basic tweet corresponds to originally authored content, and it thus identifies a single node (author).

- **Mention**: whenever a tweet of user $a$ contains a mention to user $b$, we build an edge from the author $a$ of the tweet to the mentioned account $b$.

- **Reply**: when user $a$ replies to user $b$ we build an edge from $a$ to $b$.

- **Retweet**: when user $a$ retweets another account $b$, we build an edge from $b$ to $a$.

- **Quote**: when user $a$ quotes user $b$ the edges goes from $b$ to $a$.

When processing tweets, we add a new node for users involved in aforementioned interactions whenever they are not present in the network. As a remark, a single tweet can contain simultaneously several actions and thus it can generate multiple nodes and edges. Finally, we consider edges to be weighted, where the weight corresponds to the number of times two users interacted via actions mentioned beforehand.

**Building the network of websites.** In order to investigate existing inter-connections among different disinformation websites, and to understand the nature of external sources which are usually mentioned by deceptive outlets, we searched for URLs in all articles present in our dataset, i.e. which were shared at least once on Twitter. We accordingly built a graph where each node is a distinct Top-Level Domain–the highest level in the hierarchical Domain Name System (DNS) of the Internet–and an edge is built between two nodes *a* and *b* whenever *a* has published at least an article containing an URL belonging to *b* domain; the weight of an edge corresponds to the number of shared tweets carrying an URL with an hyperlink from *a* to *b*. The final result is a directed weighted network of approximately 5k nodes and 8k edges. We used `networkx` Python package [31] to handle the network.

**Main core decomposition, centrality measures and community detection.** In our analysis we employed several techniques coming from the network science toolbox [32], namely *k*-core decomposition, community detection algorithms and centrality measures. We used `networkx` Python package to perform all the computations.

The *k*-core [33] of a graph G is the maximal connected sub-graph of G in which all vertices have degree at least *k*. Given the *k*-core, recursively removing all nodes with degree *k* allows to extract the $(k + 1)$-core; the main core is the non-empty graph with maximum value of *k*. *k*-core decomposition can be employed as to uncover influential nodes in a social network [8].

Community detection is the task of identifying *communities* in a network, i.e. dense sub-graphs which are well separated from each other [34]. In this work we consider Louvain's fast greedy algorithm [35], which is an iterative procedure that maximizes the Newman-Girvan *modularity* [36]; this measure is based on randomizations of the original graph as to check how non-random the group structure is.

A centrality measure is an indicator that allows to quantify the importance of a node in a network. In a weighted directed network we can define the *In-strength* of a node as the sum of the weights on the incoming edges, and the *Out-strength* as the sum of the weights on the outgoing edges. *Betweenness* centrality [37] instead quantifies the probability for a node to act as a bridge along the shortest path between two other nodes; it is computed as the sum of the fraction of all-pairs shortest paths that pass through the node. *PageRank* centrality [38] is traditionally used to rank webpages in search engine queries; it counts both the number and quality of links to a page to estimate the importance of a website, assuming that more important websites will likely receive more links from other websites.

## Time series analysis

In our experiments, we carried out a trend analysis of time series concerning users' activity, topics contained in disinformation articles and the number of interconnections between different outlets.

In statistics, a trend analysis refers to the task of identifying a population characteristic changing with another variable, usually time or spatial location. Trends can be increasing, decreasing, or periodic (cyclic). We used the Mann-Kendall statistical test [39, 40] as to determine whether a given time series showed a monotonic trend. The test is non-parametric and distribution-free, e.g. it does not make any assumption on the distribution of the data. The null hypothesis $H_0$, no monotonic trend, is tested against the alternative hypothesis $H_a$ that there is either an upward or downward monotonic trend, i.e. the variable consistently increases or decreases through time; the trend may or may not be linear. We used `mkt` Python package.

The multiple testing (or large-scale testing) problem occurs when observing simultaneously a set of test statistics, to decide which if any of the null hypotheses to reject [41]. In this case it is desirable to have confidence level for the whole family of simultaneous tests, e.g. requiring a

stricter significance value for each individual test. For a collection of null hypotheses we define the family-wise error rate (FWER) as the probability of making at least one false rejection, (at least one type I error). We used the classical *Bonferroni* correction to control the FWER at $\leq \alpha$ by strengthening the threshold of each individual testing, i.e. for an overall significance level $\alpha$ and $N$ simultaneous tests, we reject the individual null hypothesis at significance level $\alpha/N$.

## Limitations

As anticipated in the Introduction, our methodology has some limitations which must be considered when assessing results.

First, we remark that we investigate disinformation spreading on a single social platform (Twitter) which has not a widespread usage in Italy, specifically if compared to other social networks such as Facebook, WhatsApp and Instagram—which, however, do not exhibit good APIs for data collection.

Second, we are subject to limitations of the Twitter Streaming API; [42] indicates that the API returns at most 1% of all the tweets produced on Twitter at a given time; that source reports that once the number of tweets matching a given query exceeds 1% of the global daily volume of tweets, Twitter begins to sample the data returned to the user. In more recent documentation we found no mention of such limitation. Authors of [7, 8, 27] used the same approach as ours, and in an e-mail exchange they mentioned this Twitter policy as a potential limitation of their work. However, as the global volume of daily tweets exceeds $2 \cdot 10^8$ tweets (see https://blog.twitter.com/official/en_us/a/2011/200-million-tweets-per-day.html), most likely our data collection is not hindered by such limitation: in fact, we filter approximately $2 \cdot 10^3$ tweets per day, which are well below the 1% limit (which is roughly $2 \cdot 10^6$ tweets per day).

Third, we are collecting a specific typology of disinformation content originated from a limited set of sources, i.e. news articles published on websites which have been repeatedly flagged by journalists and fact-checkers as disinformation outlets. In line with findings from [8], we believe that we are drawing a consistent picture of Italian disinformation spreading on Twitter. However we miss photos and videos which may contain misleading or malicious content, and that can't be captured in a straightforward way. Besides, we are not verifying any of these shared items and at the same time we are not monitoring any unverified and misleading content which might be published on traditional and reliable news outlets.

Finally, for what concerns connections between disinformation outlets (see related section "Interconnections of deceptive agents") we remark that, when we observe out-going hyperlinks from Italian sources to disinformation outlets of other countries, we just show that outlets sharing disinformation news often refer to similar sources and tend to deliver similar stories; we cannot prove actual coordination between different outlets and/or countries.

## Ethics statement

We do not need ethical approval as data was publicly available and collected through Twitter Streaming API; we do not infringe Twitter terms and conditions of use. The same holds for data relative to Facebook, which was obtained using `netvizz` application in accordance with their terms of service.

## Results and discussion

### Assessing the reach of Italian disinformation

**Sources of disinformation.** To understand the reach of different disinformation outlets, we first computed the distribution of the number of articles and tweets per source. We

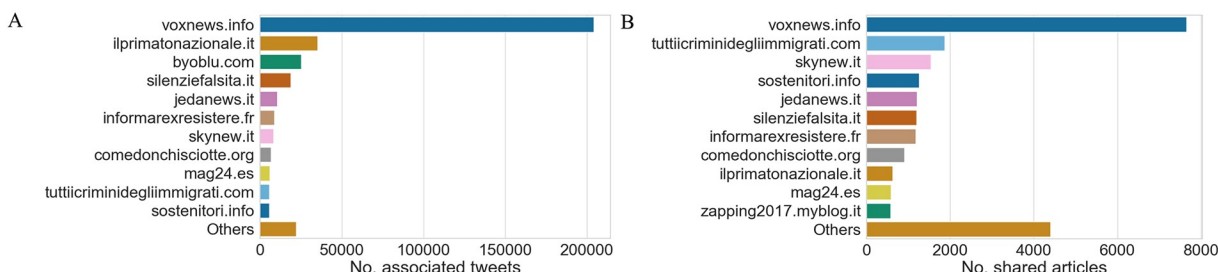

**Fig 3. A (Top).** The distribution of the total number of shared articles per website. **B (Bottom).** The distribution of the total number of associated tweets per website. We show Top-11 (which account for over 95% of the total volume of tweets), and we aggregate remaining sources as "Others".

observed, as shown in Fig 3, that a few websites dominate on the remaining ones both in terms of activity and social audience.

In particular, with approximately 200k tweets (over 50% of the total volume) and 6k articles (about 1/3 of the total number), "voxnews.info" stands out on all other sources; this outlet spreads disinformation spanning several subjects, from immigration to health-care and conspiratorial theories, and it runs campaigns against fact-checkers as well as labeling its articles with false "fact-checking" labels as to deceive readers.

Interestingly, two other uppermost prolific sources such as "skynew.it" and "tuttiicrimini-degliimmigrati.com" do not receive the same reception on the platform; the former has stopped its activity on March and the latter is literally–it translates as "All the immigrants crimes"–a repository of true, false and mixed statements about immigrants who committed crimes in Italy.

We can also recognize three websites associated to public Facebook pages that have been recently banned after the investigation of Avaaz NGO, namely "jedanews.it", "catenaumana.it" and "mag24.es", as they were "regularly spreading fake news and hate speech in Italy" violating the platform's terms of use [26].

We further computed the distribution of the daily engagement (the ratio `no.articles published/no.tweets shared` per day) per each source, noticing that a few sources exhibit a considerable number of social interactions in spite of fewer associated tweets, compared to uppermost "voxnews.info". We show the time series for the daily engagement of Top-10 sources, which account for over 95% of total tweets, in Fig 4. We can notice in particular that "byoblu.com" exhibits remarkable spikes of engagement w.r.t to a very small number of total tweets compared to other outlets, whereas "mag24.es" shows a suspiciously large number of shares in the month preceding the elections (and after the release of Avaaz report).

We excluded "ilprimatonazionale.it" from this analysis as it was added only at the end of April (we collected around 30k associated tweets and less than 1000 articles); official magazine of "CasaPound" (former) neo-fascist party–with style and agenda-setting that remind of Breit-bart News–it exhibits a daily engagement of over 200 tweets, exceeding all other websites.

As elections approached, we were interested to understand whether there were particular trends in the daily reception of different sources. Focusing on Top-10 sources (except "ilprimatonazionale.it") we performed a Mann-Kendall test to assess the presence of an upward or downward monotonic trend in the time series of (a) daily shared tweets and (b) daily engagement. Taking into account Bonferroni's correction, the test was rejected at $\alpha = 0.05/10 = 0.005$; both (a) and (b) exhibit an upward trend for "byoblu.com" alone, whereas the remaining sources are either stationary or monotonically decreasing. As this outlet strongly supported euro-skeptical positions (and often gave visibility to many Italian representatives of such

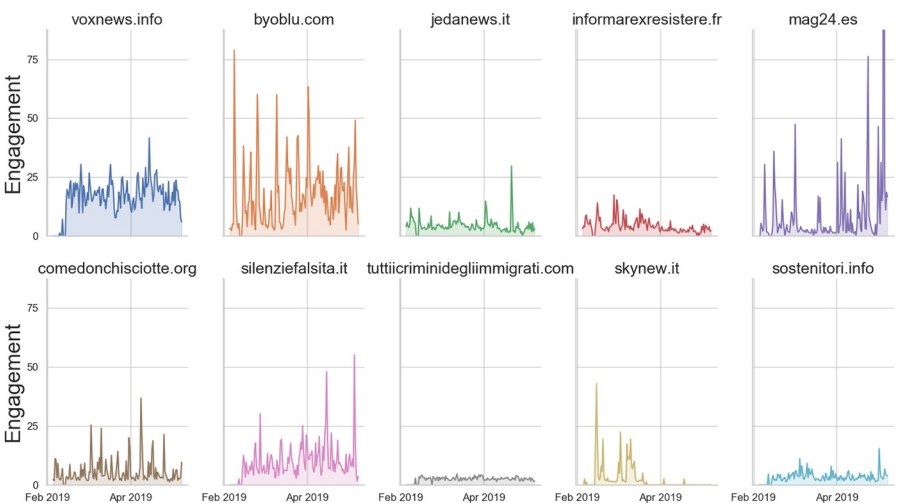

**Fig 4. Daily engagement for Top-10 sources (ranked according to the total number of shared tweets).** The Mann-Kendall test (upward trend at significance level 0.005) was accepted only for "byoblu.com".

arguments) we argue that in the run-up to the European elections its agenda became slightly more captivating for the social audience.

**User activity.** For what concerns the underlying community of users sharing disinformation, we first computed the distribution of the number of shared tweets and unique URLs shared per number of users, noticing that a restricted community of users is responsible for spreading most of the online disinformation. In fact, approximately 20% of the community ($\sim$4k users) accounts for more than 90% of total tweets ($\sim$330k), in accordance with similar findings elsewhere [8–10]. Among them, we identified accounts officially associated to 18 different outlets (we manually looked at users' profile description and usernames); they overall shared 8310 tweets.

We also distinguished five classes of users based on their generic activity, i.e. the number of shared tweets containing an URL to disinformation articles: *Rare* (about 9.5k users) with only 1 tweet; *Low* (about 8k users) with more than 1 tweet and less than 10; *Medium* (about 3k users) with a number of tweets between 11 and 100; *High* (about 500 users) with more than 100 tweets but less than 1000; *Extreme* (exactly 20 users) with more than 1000 shared tweets. About 1 user out of 5 shared more than 10 disinformation articles in five months.

As shown in Fig 5A, we can notice that a minority of very active users (the ensemble with *High* and *Extreme* activity) accounts for half of the deceptive stories that were shared, and over 3/4 of the total number of tweets was shared by less than 4 thousand users (*Medium*, *High* and *Extreme* activity).

We overall report 21,124 active (20 of which are also verified), 800 deleted, 124 protected and 112 suspended accounts. Verified accounts were altogether involved in 5761 tweets, only 18 of which in an "active" way, i.e. a verified account actually authored the tweet. We observed that they were mostly called in with the intent to mislead their followers, adding deceptive content on top of quoted statuses or replies.

Next we inspected the distribution of the number of users concerning their re-tweeting activity, i.e. the fraction of re-tweets compared to the number of pure tweets;as shown in Fig 5B this is strongly bi-modal, and it reveals that users sharing disinformation are mostly "re-tweeters": more than 60% of the accounts exhibit a re-tweeting activity larger than 0.95 and less than 30% have a re-tweeting activity smaller than 0.05. This shows that a restricted group

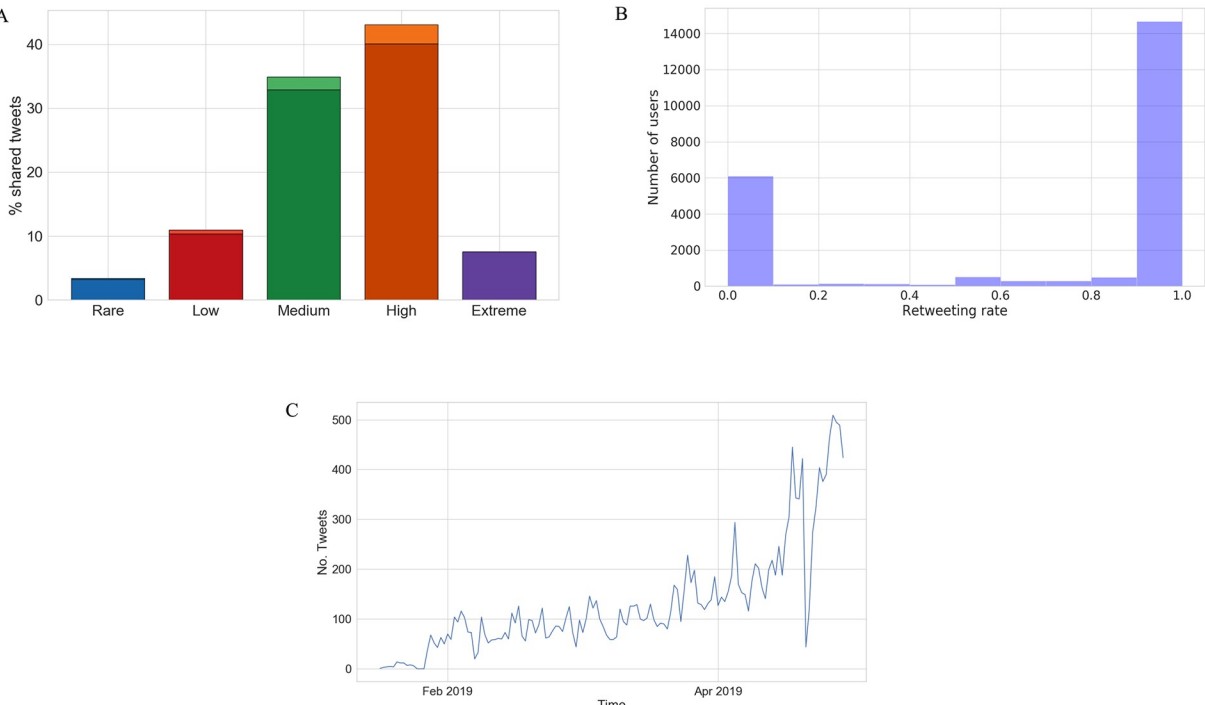

**Fig 5. A (Top).** A breakdown of the total volume of tweets according to the activity of users. Fractions of users created in the six months before the elections are indicated with lighter shades; these account respectively for 0.18% (*Rare*), 0.6% (*Low*), 2.04% (*Medium*) and 2.98% (*High*) of total tweets. **B (Center).** The distribution of the number of users per retweeting activity. **C (Bottom).** The distribution of daily tweets shared by recently created users.

of accounts is presumably responsible for conveying in the first place disinformation articles on the platform, which are propagated afterwards by the rest of the community.

We computed the distribution of some user profile features, namely the count of followers and friends, the number of statuses authored by users and the age on the social platform (in number of months passed since the creation date to May 2019). We report that users sharing disinformationtend to be quite "old" and active on the platform–with an average age of 3 years and more than a thousand authored statuses. We were able to gather information via Twitter API only for active and non-protected users.

We further inspected recently created accounts, noticing that approximately a thousand user was registered during the collection period, i.e. the last six months; they show similar distributions of aforementioned features compared to older users. Overall (see Fig 5B) they mostly pertain to active classes (*Medium* and *High*) and they account for 15% (around 18k tweets) of the total volume of tweets considered–which lowers to approximately 288k tweets excluding those authored by non-active, suspended and protected accounts. Furthermore, about a hundred exhibit abnormal activities, producing more than 10k (generic) tweets in the period preceding the elections and directly sharing more than 10 disinformation stories each. We performed a Mann-Kendall test to the time series of daily tweets shared by such users (see Fig 5C), assessing the presence of a monotonically increasing trend (at significance level $\alpha$ = 0.05). The main referenced source of disinformation is "voxnews.info" with more than 60% (circa 12k tweets) of the total number of shared stories. An activity of this kind is quite suspicious and could be further investigated as to detect the presence of "cyber-troops" (bots, cyborgs or trolls) that either attempted to drive public opinion in light of up-coming elections

(via so-called "astroturfing" campaigns [43]) or simply redirected traffic as to generate online revenue through advertisement [1–3].

## The agenda-setting of disinformation

**Theme analysis.** For what concerns the main themes covered by different disinformation outlets, relative to the resulting audience on Twitter, we based our analysis on the first level of agenda-setting theory [44], which states that news media set the public importance for objects based on the frequency in which these are mentioned and covered. In the case of disinformation news an agenda-setting effect could occur as a result of the rise in the coverage, even if some audience members are aware that news are false [45]. We focused on the prevalence of titles, which were shared at least once, as they usually pack a lot of information about their claims in simple and repetitive structures [46]; besides, the exposure (such as the presence alone of misleading titles on users' timelines) could affect ordinary beliefs and result in resistance to opposite arguments [29] and an increased perceived accuracy of the content, irrespective of its credibility [47].

We avoided automatic topic modeling algorithms [48] as they are not suitable for small texts, and we employed a dictionary-based text-analysis, an approach which is largely used for testing communication theories such as agenda setting and selective exposure in big social media data [49]. Therefore we manually compiled a list of keywords associated to five distinct topics namely: Politics/Government (PG), Immigration/Refugees (IR), Crime/Society (CS), Europe/Foreign (EF), Other (OT). Keywords were obtained with a data-driven approach, i.e. inspecting Top-500 most frequent words appearing in the titles, and taking into account relevant events that occurred in the last months. We provide Top-20 keywords for each topic in Table 1.

**Table 1. Top-20 keywords associated with each topic.**

| Politics/Government | Immigration/Refugees | Europe/Foreign | Crime/Society | Other |
|---|---|---|---|---|
| salvini | immigrati | euro | rom | video |
| italia | profughi | europa | milano | anni |
| pd | clandestini | ue | casa | contro |
| italiani | profugo | fusaro | bergoglio | foto |
| m5s | ong | diego | morti | vuole |
| italiana | porti | meluzzi | mafia | può |
| italiano | migranti | libia | bambini | vogliono |
| milioni | africani | macron | roma | parla |
| lega | immigrato | soros | donne | byoblu |
| sinistra | islamici | francia | bruciato | via |
| casapound | imam | francesi | confessa | niccolò |
| maio | seawatch | gilet | falsi | casal |
| soldi | nigeriani | gialli | bus | vero |
| guerra | nigeriana | europee | choc | ufficiale |
| cittadinanza | nigeriano | germania | figli | bufala |
| prima | islamica | tedesca | case | anti |
| raggi | africano | mondo | chiesa | sta |
| governo | stranieri | notre | famiglia | grazie |
| renzi | chiusi | dame | magistrato | casarini |
| zingaretti | sea | francese | polizia | farli |

In particular, PG refers to main political parties and state government as well as the main political themes of debate. IR includes references to immigration, refugees and hospitality whereas CS includes terms mostly referring to crime, minorities and national security. Finally EF contains direct references to European elections and foreign countries. It is worth mentioning that the most frequent keyword was "video", suggesting that a remarkable fraction of disinformation was shared as multimedia content [50].

We computed the relative presence of each topic in each article by counting the number of keywords appearing in the title and accordingly assessed their distribution across tweets over different months. We can observe in Fig 6 that the discussion was stable on controversial topics such immigration, refugees, crime and government, whereas focus on European elections and foreign affairs was quite negligible throughout the period, with only a single spike of interest at the beginning of January corresponding to the quarrel between Italian and France prime ministers. We also performed Mann-Kendall test to assess the presence of any monotonic trends in the daily distribution of different topics; we rejected the test for $\alpha = 0.05/5 = 0.01$ for IR and EF whereas we accepted it for the remaining topics, detecting the presence of an upward monotonic trend in CS and PG, and a downward monotonic trend in OT.

In the observation period, the disinformation agenda was well settled on main arguments supported by leading parties, namely "Lega" and "Movimento 5 Stelle", since 2018 general elections; this suggests that they might have profited from and directly exploited hoaxes and misleading reports as to support their populist and nationalist views (whereas "Partito Democratico" appeared among main targets of misinformation campaigns); empirical evidence for this phenomenon has been also widely reported elsewhere [23, 25]. However, the electoral outcome confirmed the decreasing trend of "Movimento 5 Stelle" electoral consensus in favor of "Lega", which was rewarded with an unprecedented success.

Differently from 2018 [23] we in fact observed one main cited leader: Matteo Salvini ("Lega" party). This is consistent with a recent report on online hate speech [51], contributed by Amnesty International, which has shown that his activity (and reception) on Twitter and Facebook is 5 times higher than Luigi Di Maio (leader of "Movimento 5 Stelle"); not

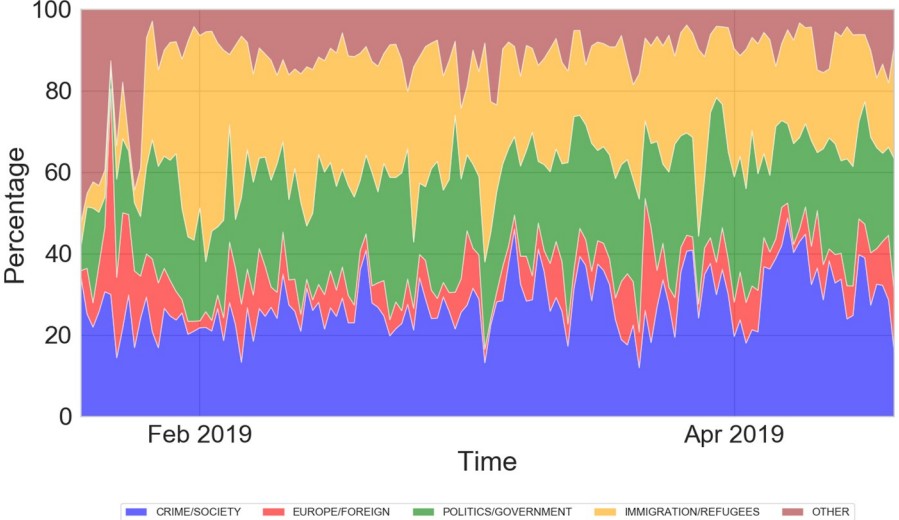

**Fig 6. A stacked-area chart showing the distribution of different topics over the collection period.** The daily coverage on themes related to Immigration/Refugees and Europe/Foreign is stationary, whereas focus on subjects related to Crime/Society and Politics/Government is monotonically increasing towards the elections (end of May 2019).

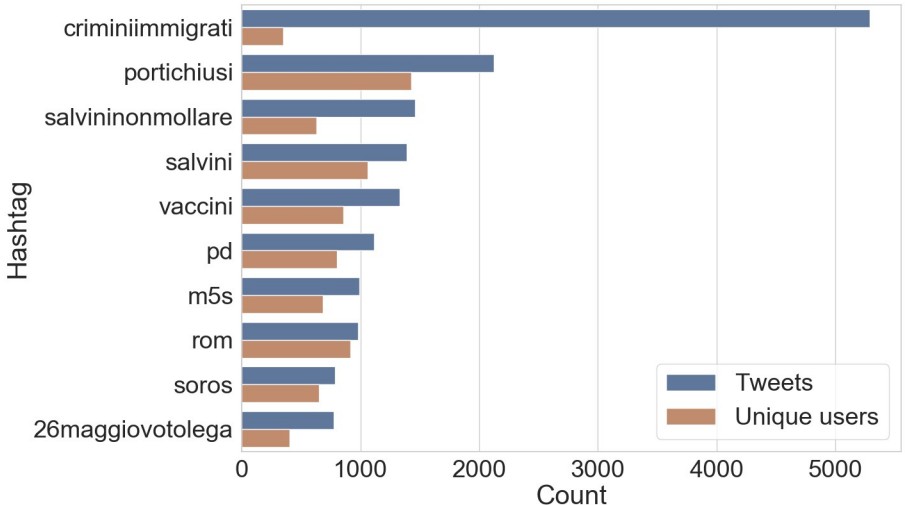

**Fig 7. Top-10 hashtags per number of shared tweets (blue) and unique users (orange).**

surprisingly, his main agenda focuses (negatively) on immigration, refugees and Islam (which generated most of online interactions in 2018 [23]), which are also the main objects of hate speech and controversy in online conversations of Italian political representatives overall.

It appears that mainstream news actually disregarded European elections in the months preceding them, focusing on arguments of national debate [52]; this trend was also observed in other European countries according to FactCheckEU [53], claiming that misinformation-was not prominent in online conversations mainly because European elections are not particularly polarized and are seen as less important compared to national elections. We believe that this might have affected the agenda of disinformation outlets, which are in general susceptible to traditional media coverage [54], thus explaining the focus on different targets in their deceptive strategies.

**Usage of hashtags.** Among most relevant hashtags shared along with tweets–in terms of number of tweets and unique users who used them (see Fig 7)–a few indicate main political parties (cf. "m5s", "pd", "lega") and others convey supporting messages for precise factions, mostly "Lega" (cf. "salvininonmollare", "26maggiovotolega"); some hashtags manifest instead active engagement in public debates which ignited on polarizing and controversial topics (such as immigrants hospitality, vaccines, the Romani community and George Soros). We also found explicit references to (former) far-right party "CasaPound" and the associated "Altaforte" publishing house, as well as some disinformation websites (with a remarkable polarization on "criminiimmigrati" which was shared more than 5000 times by only a few hundred accounts).

We also extracted hashtags directly embedded in the profile description of users collected in our data, for which we provide a cloud of words in Fig 8. The majority of them expresses extreme positions in matter of Europe and immigration: beside explicit references to "Lega" and "Movimento 5 Stelle", we primarily notice euro-skeptical (cf. "italexit", "noue"), anti-Islam (cf. "noislam") and anti-immigration positions (cf. "noiussoli", "chiudiamo i porti") and, surprisingly enough, also a few (alleged) Trump followers (cf. "maga" and "kag"). The latter finding is odd but somehow reflects the vicinity of Matteo Salvini and Donald Trump on several political matters (such as refugees and national security). On the other hand, we also notice "facciamorete", which refers to a Twitter grassroots anti-fascist and anti-racist

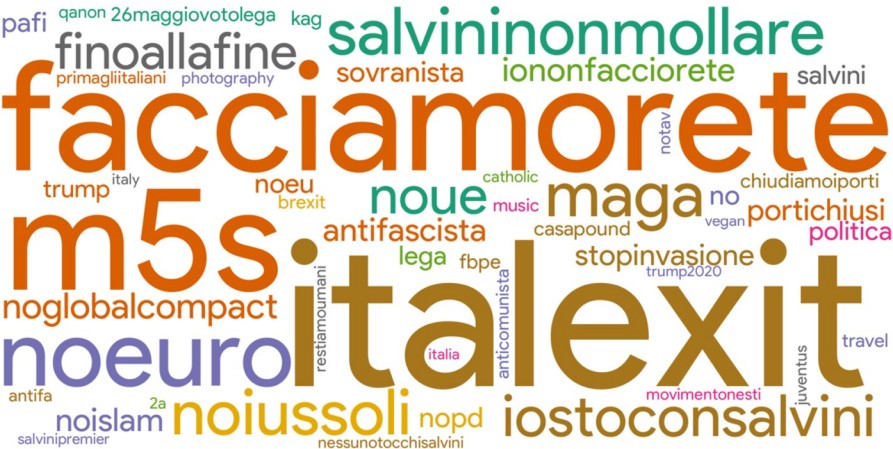

**Fig 8. The cloud of words for Top-50 most frequent hashtags embedded in the users' profile description.**

movement that was born on December 2018, as a reaction to the recent policies in matter of immigration and national security of the Italian establishment.

### Principal spreaders of disinformation

**Central users in the main core.** In order to identify most influential nodes in the diffusion network, we computed the value of several centrality measures for each account. We show in Table 2 the list of Top-10 users according to each centrality measure, and we also indicate whether they belong or not to the main K-core of the network [33]; this corresponds to the sub-graph of neighboring nodes with degree greater or equal than $k = 47$, which is shown in Fig 9. We color nodes according to the communities identified by the Louvain modularity-based community algorithm [35] run on the original diffusion network (over 20k nodes and 100k edges).

Although we expect centrality measures to display small differences in their ranking, we can notice that the majority of nodes with highest values of In-Strength, Out-Strength and Betweenness centralities also belong to the main K-core of the network; the same does not hold for users which have a large PageRank centrality value. A few users strike the eye:

**Table 2. List of Top-10 users according to different centrality measures, namely In-strength, Out-Strength, Betweenness and PageRank; we indicate with a cross nodes that do not belong to the main K-core ($k = 47$) of the network.**

| Rank | In-Strength | Out-Strength | Betweenness | PageRank |
|---|---|---|---|---|
| 1 | napolinordsud ✕ | Filomen30847137 | IlPrimatoN | IlPrimatoN |
| 2 | RobertoPer1964 | POPOLOdiTWlTTER | matteosalvinimi | matteosalvinimi |
| 3 | razorblack66 | laperlaneranera | Filomen30847137 | Sostenitori1 ✕ |
| 4 | polizianuovanaz ✕ | byoblu | byoblu | armidmar |
| 5 | Giulia46489464 | IlPrimatoN | a_meluzzi | Conox_it ✕ |
| 6 | geokawa | petra_romano | AdryWebber | lauraboldrini ✕ |
| 7 | Gianmar26145917 | araldoiustitia | claudioerpiu | pdnetwork ✕ |
| 8 | pasqualedimaria ✕ | max_ronchi | razorblack66 | libreidee ✕ |
| 9 | il_brigante07 | Fabio38437290 | armidmar | byoblu |
| 10 | AngelaAnpoche | claudioerpiu | Sostenitori1 ✕ | Pontifex_it ✕ |

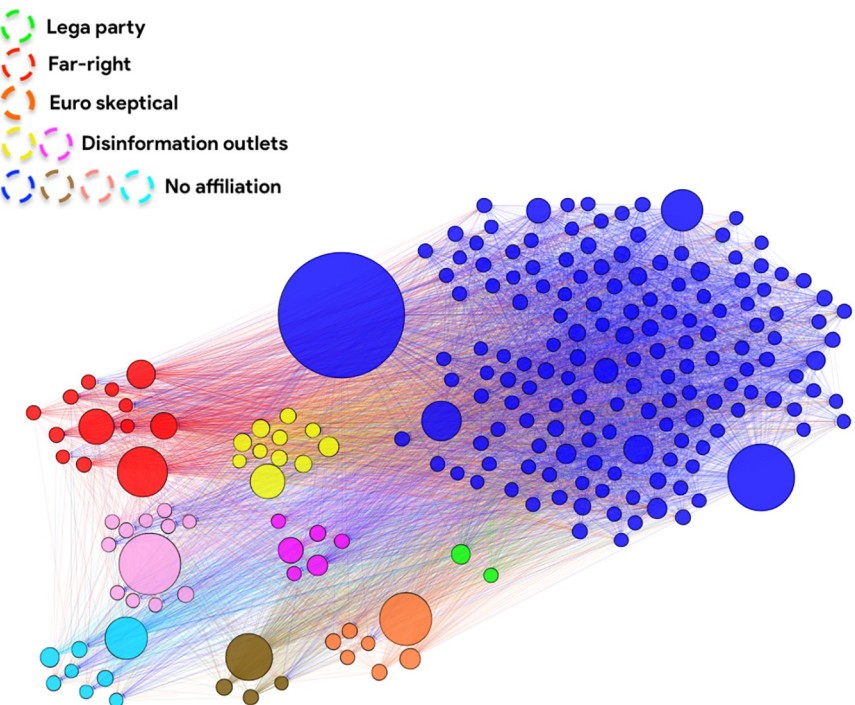

**Fig 9. The main K-core ($k = 47$) of the re-tweeting diffusion network.** Colors correspond to different communities identified with the Louvain's algorithm. Node size depends on the total Strength (In + Out) and edge color is determined by the source node.

1. matteosalvinimi is Matteo Salvini,leader of the far-right wing "Lega" party; he is not an active spreader of disinformation, being responsible for just one (true) story coming from disinformation outlet "lettoquotidiano.com" (available at https://twitter.com/matteosalvinimi/status/1102654128944308225), which was shared over 1800 times. He is generally passively involved in deceptive strategies of malicious users who attempt to "lure" his followers by attaching disinformation links in replies/re-tweets/mentions to his account.

2. a_meluzzi is Alessandro Meluzzi, a former representative of centre-right wing "Forza Italia" party (whose leader is Silvio Berlusconi); he is a well-known supporter of conspiracy theories and a very active user in the disinformation network, with approximately 400 deceptive stories shared overall.

3. Accounts associated to disinformation outlets, namely IlPrimatoN with "ilprimatonazionale.it", byoblu with "byoblu.com", libreidee with "libreidee.org", Sostenitori1 with "sostenitori.info" and Conox_it with "conoscenzealconfine.it".

A manual inspection revealed that most of the influential users are indeed actively involved in the spread of disinformation, with the only exception of matteosalvinimi who is rather manipulated by other users, via mentions/retweets/replies, as to mislead his huge community of followers (more than 2 millions). The story shared by Matteo Salvini underlinesa common strategy of disinformation outlets identified in this analysis: they often publish simple true and factual news as to bait users and expose them to other harmful and misleading content present on the same website.

Besides, we notice in the ranking a few users who are (or have been in the past) target of several disinformation campaigns, such as `lauraboldrini` (Laura Boldrini), `pdnetwork` ("Partito Democratico" party) and `Pontifex_it` (Papa Francesco). We also report a suspended account (`polizianuovanaz`), a protected one (`Giulia46489464`) and a deleted user (`pasqualedimaria`).

In addition, we investigated communities of users in the main K-core–which contains 218 nodes (see Fig 9)–and we noticed systematic interactions between distinct accounts. We manually inspected usernames, most frequent hashtags and referenced sources, deriving the following qualitative characterizations:

1. the **Green** community corresponds to "Lega" party official accounts: `matteosalvinimi` and `legasalvini`, whereas the third account, `noipersalvini`, belongs to the same community but does not appear in the core.

2. the **Red** community represents Italian far-right supporters, with several representatives of CasaPound (former) party (including his secretary `distefanoTW` who does not appear in the core), who obviously refer to "ilprimatonazionale.it" news outlet.

3. the **Yellow** community is strongly associated to two disinformation outlets, namely "silenziefalsita.it" (`SilenzieFalsita`) and "jedanews.it" (`jedasupport`); the latter was one of the pages identified in Avaaz report [26] and deleted by Facebook.

4. the **Orange** community is associated to the euro-skeptical and conspiratory outlet "byoblu.com" (`byoblu`), and it also features Antonio Maria Rinaldi (`a_rinaldi`), a well-known euro-skeptic economist who has just been elected with "Lega" in the European Parliament.

5. the **Purple** community corresponds to the community associated to "tuttiicriminidegliimmigrati.com" (`TuttICrimin`) disinformation outlet.

6. the remaining **black** (`Filomen30847137`), **Light-blue** (`araldoiustitia`) and **Brown** communities (`petra_romano`) represent different groups of very active "loose cannons" who do not exhibit a clear affiliation.

Eventually, we employed Botometer algorithm [55] as to detect the presence of social bots among users in the main core of the network. We set a threshold of 50% on the Complete Automation Probability (CAP)–i.e. the probability of an account to be completely automated–which, according to the authors, is a more conservative measure that takes into account an estimate of the overall presence of bots on the network; besides, we computed the CAP value based on the language independent features only, as the model includes also some features conceived for English-language users. We only detected two bot-like accounts, namely `simonemassetti` and `jedanews`, respectively with probabilities 58% and 64%, that belong to the same Purple community. A manual check confirmed that the former habitually shares random news content (also mainstream news) in an automatic flavour whereas the latter is the official spammer account of "jedanews.it" disinformation outlet. We argue that the impact of automated accounts in the diffusion of malicious information is quite negligible compared to findings reported in [8], where about 25% of accounts in the main core of the US disinformation diffusion network were classified as bots.

**Dismantling the disinformation network on Twitter.**  Similar to [8], we performed an exercise of network dismantling analysis using different centrality measures, as to investigate possible intervention strategies that could prevent disinformation from spreading with the greatest effectiveness.

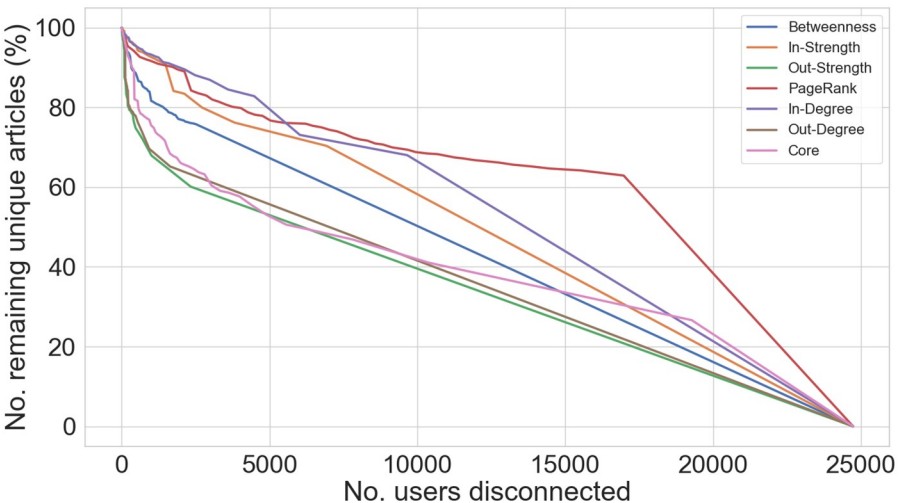

**Fig 10. Results of different network dismantling strategies w.r.t to remaining unique disinformation articles in the network.** The x-axis indicates the number of disconnected accounts and the y-axis the fraction of remaining items in the network.

We first ranked nodes in decreasing order w.r.t to each metric, plus the core number–the largest $k$ for which the node is present in the corresponding $k$-core–and the In and Out-degree, which exhibited the same Top 10 ranking as their weighted formulation (Strengths), but they do entail different results at dismantling the network. Next we delete them one by one while tracking the resulting fraction of remaining edges, tweets and unique articles in the network.

We observed that eliminating a few hundred nodes with largest values of Out-Degree promptly disconnects the network;in fact these users alone account for 90% of the total number of interactions between users. For what concerns the number of tweets sharing disinformation articles, the best strategy would be to target users with largest values of In-Strength who, according to our network representation, are likely to be users with a high re-tweeting activity; in fact, confirming previous observations, a few thousand nodes account for more than 75% of the total number of tweets shared in the five months before the elections. However, as shown in Fig 10, it is more challenging to prevent users to be exposed from even a tiny fraction of disinformation articles, as the network exhibits an almost linear relationship between the number of users disconnected and the corresponding number of remaining stories; as such the spread of malicious information would be completely prevented only blocking the entire network.

### Interconnections of deceptive agents

To investigate existing connections between different disinformation outlets and other external sources, we first analyzed the network of websites with a core decomposition [33], obtaining a main core ($k = 14$) which contains 35 nodes as a result of over75,000 external re-directions via hyperlinks (shown in Fig 11A). Over 99% of the articles includes a hyperlink in the body. We may first notice frequent connections between distinct disinformation outlets, suggesting the presence of shared agendas and presumably coordinated deceptive tactics, as well as frequent mentions to reputable news websites; among them we distinguish "IlFatto-Quotidiano", which is a historical supporter of "Movimento 5 Stelle", and conservative outlets such as "IlGiornale" and "LiberoQuotidiano" which lean instead towards "Lega". We also observe that most of the external re-directions point to social networks (Facebook and Twitter) and video sharing websites (Youtube); this is no wonder given that disinformation is often

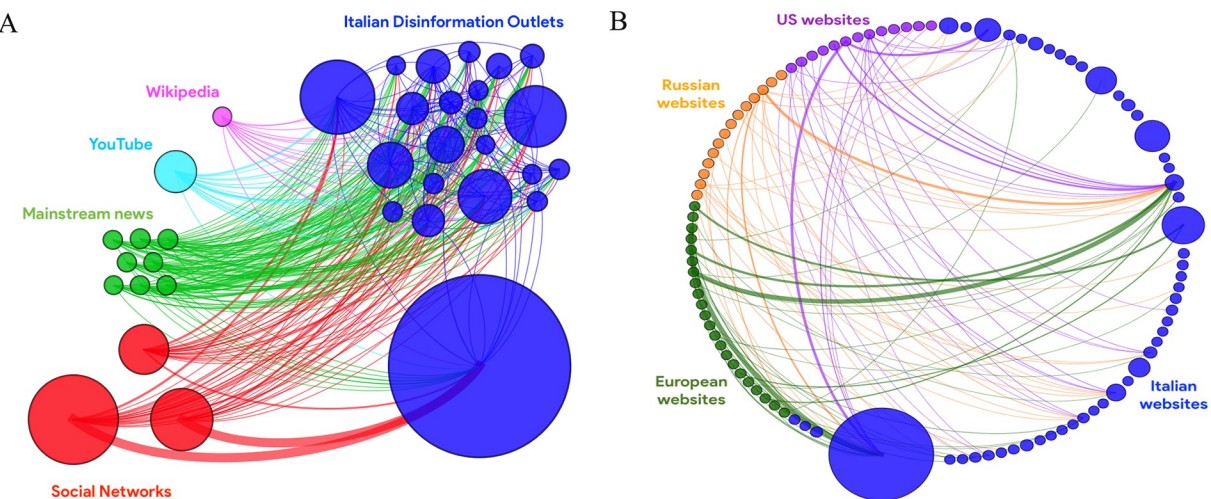

**Fig 11. Two different views of the network of websites; the size of each node is adjusted w.r.t to the Out-strength, the color of edges is determined by the target node and the thickness depends on the weight (i.e. the number of shared tweets containing an article with that hyperlink). A (Left).** The main core of the network ($k = 14$); blue nodes are Italian disinformation websites, green ones are Italian traditional news outlets, red nodes are social networks, the sky-blue node is a video sharing website and the pink one is an online encyclopedia. **B (Right).** The sub-graph of Russian (orange), EU (olive green), US (violet) and Italian (blue) disinformation outlets.

shared on social networks as multimedia content [1, 3]. In addition, we inspected nodes with the largest number of incoming edges (In-degree) in the original network, discovering among uppermost 20 nodes a few misleading reports originated on dubious websites (such as "neoingegneria.com"), flagged by fact-checkers but that were not included in any blacklist. We believe that a more detailed network analysis could reveal additional relevant connections and we leave it for future research.

Furthermore, we focused on the sub-graph composed of three particular classes of nodes, namely Russian (RU) sources, EU/US disinformation websites and our list of Italian (IT) outlets; we manually identified notable Russian sources ("RussiaToday" and "SputnikNews" networks) and we resorted to notable blacklists to spotlight other EU/US disinformation websites–namely "opensources.co", "décodex.fr", the list compiled by Hoaxy [27] and references to junk news in latest data memos by COMPROP research group [14, 16–18].

The resulting bipartite network–we filtered out intra-edges between IT sources to better visualize connections with the "outside" world–contains over 60 foreign websites (RU, US and EU) and it is shown in Fig 11B.

We observe a considerable number of external connections (over 500 distinct hyperlinks present in articles shared more than 5 thousand times) with other countries sources, which were primarily included within "voxnews.info", "ilprimatonazionale.it" and "jedanews.it". Among foreign sources we encounter several well-known US sources ("breitbart.com", "naturalnews.com" and "infowars.com" to mention a few) as well as RU ("rt.com", "sputniknews.com" and associated networks in several countries), but we also find interesting connections with notable disinformation outlets from France ("fdesouche.com" and "breizh-info.com"), Germany ("tagesstimme.com"), Spain ("latribunadeespana.com") and even Sweden ("nyheteridag.se" and "samnytt.se"). Besides, a manual inspection of a few articles revealed that stories often originated in one country were immediately translated and promoted from outlets in different countries (see Fig 12). Such findings suggest the existence ofinter-connected deceptive strategies which span across several countries, consistently with claims in latest report by Avaaz [26] which revealed the existence of a network of far-right and anti-EU websites, leading

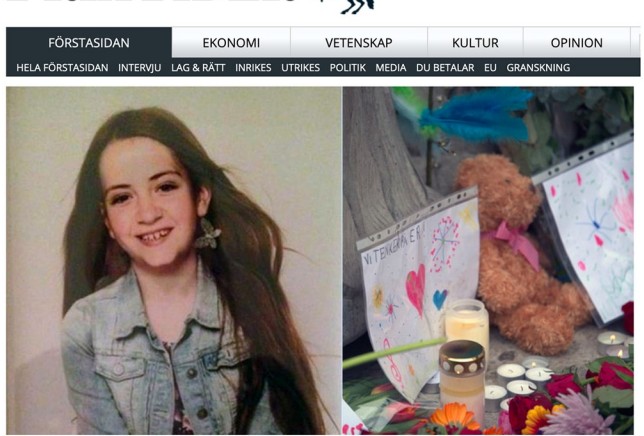

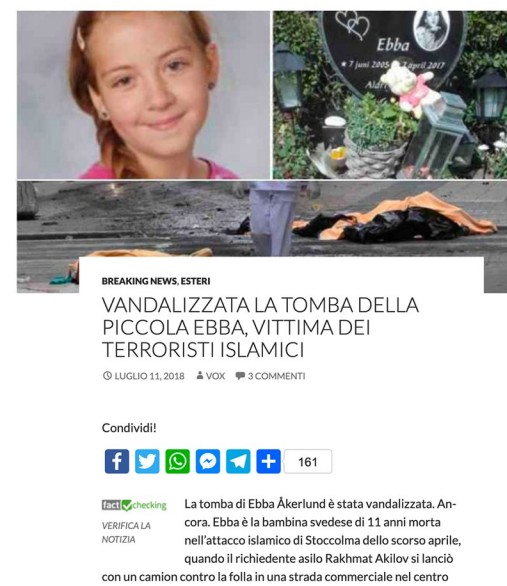

**Fig 12. An example of disinformation story who was published on a Swedish website ("friatider.se") and then reported by an Italian outlet ("voxnews.info").** Interestingly, this news is old (July 2018) but it was diffused again in the first months of 2019.

to the shutdown of hundreds of Facebook pages with more than 500 million views just ahead of the elections. Far-right disinformation tactics comprised the massive usage of fake and duplicate accounts, recycling followers and bait and switch of pages covering topics of popular interest (e.g. sport, fitness, beauty).

It is interesting that Facebook decided on the basis of external insights to shutdown pages delivering misleading content and hate speech; differently from the recent past [3, 7, 8] it might signal that social media are more willing to take action against the spread of deceptive information in coordination with findings from third-party researchers. Nevertheless, we argue that closing malicious pages is not sufficient and more proactive strategies should be followed [3, 26].

In order to check the relevance of inter-connections with websites of different countries, we applied a simple degree preserving randomization [56] to the network depicted in Fig 11B and tested whether the percentages of links respectively towards EU, US and RU were significantly different from the mean value observed in the random ensemble (obtained re-wiring the network for 1000 times). We thus performed a Z-test at $\alpha = 0.05/3$, rejecting the null hypothesis in all cases; in particular the number of RU and US connections are higher than expected whereas the number of EU connections is lower.

Finally, we performed a Mann-Kendall test to see whether there was an increasing trend, towards the elections, in the number of external connections with US and RU disinformation websites; we rejected it at $\alpha = 0.05/2 = 0.0025$.

## Conclusions

We studied the reach of Italian disinformation on Twitter for a period of five months immediately preceding the European elections (**RQ1**) by analyzing the content production of websites producing disinformation, and the characteristics of users sharing malicious items on the social platform. Overall, thousands of articles–which included hoaxes, propaganda, hyper-partisan and conspiratorial news–were shared in the period preceding the elections. We observed that a few outlets accounted for most of the deceptive information circulating on Twitter;

among them, we also encountered a few websites which were recently banned from Facebook after violating the platform's terms of use. We identified a heterogeneous yet limited community of thousands of users who were responsible for sharing disinformation. The majority of the accounts (more than 75%) occasionally engaged with malicious content, sharing less than 10 stories each, whereas only a few hundred accounts were responsible for (the spreading) of thousands of articles (see Fig 5).

We singled out the most debated topics of disinformation **(RQ2)** by inspecting news items and Twitter hashtags. We observed that they mostly concern polarizing and controversial arguments of the local political debate such as immigration, crime and national safety, whereas discussion around the topics of Europe global management had a negligible presence throughout the collection period; the lack of European topics was also reported in theagenda of mainstream media.

Then we identified the most influential accounts in the diffusion network resulting from users sharing disinformation articles on Twitter (**RQ3**), so as to detect the presence of active groups with precise political affiliations. We discovered strong ties with the Italian far-right and conservative community, in particular with "Lega" party, as most of the users manifested explicit support to the party agenda through the use of keywords and hashtags. Besides, a common deceptive strategy was to passively involve his leader Matteo Salvini via mentions, quotes and replies as to potentially mislead his audience of million of followers. We found limited evidence of bot activity in the main core, and we observed that disabling a limited number of central users in the network wouldconsiderably reduce the spread of disinformation circulating on Twitter, but it would immediately raise censorship concerns.

Finally, we investigated inter-connections within different deceptive agents (**RQ4**), thereby observing that they repeatedly linked to each other websites during the period preceding the elections. Moreover we discovered many cases where the same (or similar) stories were shared in different languages across different European countries.

This analysis confirms that disinformation is present on Twitter and that its spread shows some peculiarities in terms ofthemes being discussed and of political affiliation of the key members of the information spreading community. We are aware that disinformation news in Italy have a higher share on Facebook than Twitter and that the use of Twitter in Italy as a social channel is limited compared to other social platforms such as Facebook, WhatsApp or Instagram. Therefore similar studies on other social media platforms will be needed and beneficial to our understanding of the spread of disinformation.

## Acknowledgments

The authors are very grateful to PagellaPolitica and Butac for providing support to build the list of disinformation outlets, to Hoaxy support team at Bloomington Indiana University for providing insights and support on the collection of data, to Carlo Piccardi for providing suggestions related to network analysis, and to anonymous reviewers for their comments and suggestions which improved the quality of the final manuscript.

## Author Contributions

**Conceptualization:** Francesco Pierri.

**Data curation:** Francesco Pierri, Alessandro Artoni.

**Formal analysis:** Francesco Pierri.

**Investigation:** Francesco Pierri, Alessandro Artoni.

**Methodology:** Francesco Pierri.

**Resources:** Alessandro Artoni.

**Software:** Alessandro Artoni.

**Supervision:** Stefano Ceri.

**Validation:** Stefano Ceri.

**Visualization:** Francesco Pierri, Alessandro Artoni.

**Writing – original draft:** Francesco Pierri.

**Writing – review & editing:** Francesco Pierri, Stefano Ceri.

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
