## [Decision Letter · Decision Letter 0]

10 Sep 2019

PONE-D-19-20362

Investigating Italian disinformation spreading on Twitter in the context of 2019 European elections

PLOS ONE

Dear Mr. Pierri,

Thank you for submitting your manuscript to PLOS ONE. After careful consideration, we feel that it has merit but does not fully meet PLOS ONE’s publication criteria as it currently stands. Therefore, we invite you to submit a revised version of the manuscript that addresses the points raised during the review process.

Reviewers have indeed expressed some concerns about this paper and pointed out several weaknesses that need to be addressed before publication. In particular, Reviewer 3 observed a discrepancy in numbers between the manuscript and the .pkl file. Moreover, he noticed that the uploaded dataset does not appear to be clean, perhaps due to scraping errors. I would strongly suggest to address this issue, since an error in data collection would seriously affect the results of the paper. Important concerns were also raised w.r.t. how the topic analysis was performed and how the retweeting diffusion network was created. If you are willing to revise the manuscript, please address all reviewers’ comments and clearly indicate how you have addressed them.

We would appreciate receiving your revised manuscript by Oct 25 2019 11:59PM. To enhance the reproducibility of your results, we recommend that if applicable you deposit your laboratory protocols in protocols.io, where a protocol can be assigned its own identifier (DOI) such that it can be cited independently in the future. For instructions see: http://journals.plos.org/plosone/s/submission-guidelines#loc-laboratory-protocols

We look forward to receiving your revised manuscript.

Kind regards,

Fabiana Zollo, Ph.D.

Academic Editor

PLOS ONE

1. In your data collection section, please include a statement clarifying that you obtained the data from Twitter and Facebook in accordance with their terms of service. Thank you for your attention to this query.

Reviewers' comments:

Reviewer's Responses to Questions

**Comments to the Author**

1. Is the manuscript technically sound, and do the data support the conclusions?

Reviewer #1: Yes

Reviewer #2: Yes

Reviewer #3: Partly

2. Has the statistical analysis been performed appropriately and rigorously? 

Reviewer #1: Yes

Reviewer #2: Yes

Reviewer #3: I Don't Know

3. Have the authors made all data underlying the findings in their manuscript fully available?

Reviewer #1: Yes

Reviewer #2: Yes

Reviewer #3: Yes

4. Is the manuscript presented in an intelligible fashion and written in standard English?

Reviewer #1: Yes

Reviewer #2: Yes

Reviewer #3: Yes

5. Review Comments to the Author

Reviewer #1: The manuscript offers an intersting and timely perspective on the public debate in the Italian Twittersphere during the last European Parliamentary Elections.

The few available academic contributions so far seem to agree on the limited scope of online disinformation during the campaign; social bots and botnets also seem to have played a limited role. In this respect, the work under examination appears to be in line with other studies on the same topic.

The authors adopt a balanced and informed stance. The manuscript does not offer - nor promise, indeed - an innovative approach to the study of online disinformation, yet it can usefully contribute to the interdisciplinary analysis of political participation during a main event such as the EP elections.

The technical analyses have been carried out appropriately and clearly described.

The manuscript is well organized and written clearly enough to be accessible to non specialists.

I have three comments on:

1) data collection

2) bot detection

3) style - some suggestions to improve the readability of the manuscript (typos, false friends, possible rephrasing).

1) The authors state (line 109) that they gathered "100% of shared tweets matching the defined query (see next).". I wonder whether they applied for and got access to the new Premium APIs, or adopted the standard Streaming API - as declared above (line 106). I think this point might need some clarification.

2) The authors are well aware that social bots can have a direct role in spreading disinformation, therefore adopting state-of-the-art bot detection techniques might be beneficial for a thorough analysis of the pollution of the online debate. Recent and promising techniques are oriented towards group (rather than account-by-account) analysis and unsupervised (rather than supervised) approaches; among them: Mazza M. et al. (2019), RTBust, Cresci S. et al. (2017), Social Fingerprinting. I am not implying the authors shoud adopt one of them, but they might consider justifying the use of the Botometer algorithm compared to other approaches.

3) I will list my suggestions:

- line 21: replace with "which seldom "affected/had a limited impact on online discussions..."

- line 28: As people "are" more and more...

- line 35: broader than "reliable news" (rather than "truth")

- line 43 (+ lines 48, 277 and 235): replace "for what concerns" with "as for/as far as...is concerned/with regard to"

- line 59: replace "contemporary" with "very recent"

- line 66: inflammatory content...about controversial themes such as... (inserire "about", eliminare "of debate")

- line 70: replace "pursue" with "consider/focus on/carry out our research on" a consolidated setting...

- Fig. 1 (above line 90): replace "annotate it" with "annotated them"

- line 149: replace "observed by other sources" with "shown in other works"

- line 173: replace "within" with "among" different disinformation websites...

- line 217: replace "referred to" with "adopted/used" (the) mkt Python package

- line 220: In this case (it) is desirable to have (a) confidence level

- line 312: replace "tendentiously" with "tend to be" quite...

- line 389: replace "struggled to dominate" with "did not dominate/were not prominent in" online conversations

- line 390: replace "are not very interesting compared to" with "are seen as less important than" national elections

- line 416: replace "actual" with "current"

- line 417: replace "Relevant" with "Principal/Main" spreaders of disinformation

- line 427: replace "Despite" with "Although", and "agree" with "coincide"

- line 429: replace "belongs also" with "also belong"

- line 441: replace "affiliated to" with "whose leader is"

- line 453: replace "as to" with "in order to"

- line 459: "repetitively" with "continuously"

- line 461: "actual" with "current"

- lines 463-4: replace "statements in favor of immigration" with "statements expressing solidarity with/showing empathy for migrants"

- line 515: might be rephrased as "in fact these users alone account for..."

- line 524-5: might be rephrased as "as such the spread of malicious information would be completely prevented only blocking/removing the entire network"

- line 583: replace "six" with "five" months (at least so it is stated in the abstract and in the Data Collection paragraph)

- line 591: "recognized" with "identified/characterized"

- lines 594-5: might be rephrased as "only a few hundred accounts were responsible for (the spreading) of thousands of articles" (it might be convenient to specify the figures)

- line 596: replace "we then studied" with "we singled out"

- line 602: invert as "Then we identified"

- lines 619-20: might be rephased as "its spread shows some peculiarities, in terms of..."

- line 621: replace "Note, however, that.." with "We are aware that the main..."

- line 623: "small" with "limited" and "sources" with "platforms".

- lines 624-5: (full stop after "Instagram") might be reprhased as: "Therefore similar studies on other social media platforms will be needed and beneficial to our understanding of the spread of disinformation."

Reviewer #2: The paper discusses some results related to the spreading of false news in the period preceding the European elections. Overall, the paper is interesting, clear and relatively sound in the methods. I recommend the paper for publication after some minor issues will be fixed.

Reviewer #3: See attachment. (Apparently, it's not possible to submit if this box has less than 100 characters, so here's some text to get there.)

6. PLOS authors have the option to publish the peer review history of their article (what does this mean?). If published, this will include your full peer review and any attached files.

Reviewer #1: No

Reviewer #2: No

Reviewer #3: No

---

## [Author Response · Author response to Decision Letter 0]

17 Sep 2019

(see attached file "response_reviews_final.docx" for a better view of comments/answers)

---

## [Decision Letter · Decision Letter 1]

21 Oct 2019

PONE-D-19-20362R1

Investigating Italian disinformation spreading on Twitter in the context of 2019 European elections

PLOS ONE

Dear Mr. Pierri,

Thank you for submitting your manuscript to PLOS ONE. After careful consideration, we feel that it has merit but does not fully meet PLOS ONE’s publication criteria as it currently stands. Therefore, we invite you to submit a revised version of the manuscript that addresses the points raised during the review process.

I would invite the authors to address all the points raised by the reviewers. In particular, a statistical validation of the analysis on the connection between Italy and Russia is needed, as well as a method reference for topic modelling. Overall, the authors should work to improve the clarity of the paper (both reviewers noticed that the language is often inconsistent) and to proof-read before resubmission. I would also invite the authors to add a Limitations section to the manuscript and to discuss here the qualitative process adopted for the recognition of influential users, along with all the limitations emerged during the revision process. As far as the Twitter API, I would suggest the authors to pay attention to the reviewer’s comments and to discuss the issue in the same section, if appropriate.

We would appreciate receiving your revised manuscript by Dec 05 2019 11:59PM. To enhance the reproducibility of your results, we recommend that if applicable you deposit your laboratory protocols in protocols.io, where a protocol can be assigned its own identifier (DOI) such that it can be cited independently in the future. For instructions see: http://journals.plos.org/plosone/s/submission-guidelines#loc-laboratory-protocols

We look forward to receiving your revised manuscript.

Kind regards,

Fabiana Zollo, Ph.D.

Academic Editor

PLOS ONE

Reviewers' comments:

Reviewer's Responses to Questions

**Comments to the Author**

1. If the authors have adequately addressed your comments raised in a previous round of review and you feel that this manuscript is now acceptable for publication, you may indicate that here to bypass the “Comments to the Author” section, enter your conflict of interest statement in the “Confidential to Editor” section, and submit your "Accept" recommendation.

Reviewer #2: (No Response)

Reviewer #3: (No Response)

2. Is the manuscript technically sound, and do the data support the conclusions?

Reviewer #2: Yes

Reviewer #3: Yes

3. Has the statistical analysis been performed appropriately and rigorously? 

Reviewer #2: N/A

Reviewer #3: I Don't Know

4. Have the authors made all data underlying the findings in their manuscript fully available?

Reviewer #2: Yes

Reviewer #3: Yes

5. Is the manuscript presented in an intelligible fashion and written in standard English?

Reviewer #2: Yes

Reviewer #3: Yes

6. Review Comments to the Author

Reviewer #2: The authors improved the paper and (shortly) addressed my concerns. Reading again the paper I noted very few points that captured my attention and that need further clarification.

Reviewer #3: I'm pretty much fine with the current version, but I’ve made comments handwriting in the document (second half of the pdf.) See attached. (And sorry for my handwriting. :)

-- I checked Shao et al 2008 for any mention of the claim that 100% of tweets would be captured, and found nothing. What page/line are you referring to? I then looked at Twitter API documentation on how much the free streaming API captures, and only came up with something like 1%. There’s no mention that this would be different just because you’re tracking URLs. There’s also plenty of literature on the limitations of using the Free Streaming API, for instance that the selection of tweets is not a random sample. I would like a reference to the documentation where it’s stated that this in fact captures 100% of tweets. Or if it in fact does not capture 100%, a mention of this issue in the limitation section. (If it does not capture 100% of tweets, this likely impacts other parts of the text making claims of the prevalence of misinformation tweets.)

-- The new text needs to be checked for spelling and grammar. See comments in pdf.

-- Check reference list. Some entries lack year/journal/etc.

-- Check that terminology is consistent: now it uses theme/topic, fake/false/junk news, etc.

-- The method of the theme/topic modeling is still a bit unclear to me. E.g. how are the list of topic words summed up? Adding a method reference to someone else using the same method would be good.

-- The "targets of misinformation" section method is now clarified as "we recognized a few influential users". Is this really the method? I'm all for qualitative approaches, but just looking for things that you recognize seems a bit strong? Also, is the section really necessary - I'm not sure what it adds?

-- The pp22 section is still called "coordinated strategies of deception" even though the rest of the text is changed. Do change this.

7. PLOS authors have the option to publish the peer review history of their article (what does this mean?). If published, this will include your full peer review and any attached files.

Reviewer #2: No

Reviewer #3: No

---

## [Decision Letter · Decision Letter 2]

28 Nov 2019

PONE-D-19-20362R2

Investigating Italian disinformation spreading on Twitter in the context of 2019 European elections

PLOS ONE

Dear Mr. Pierri,

Thank you for submitting your manuscript to PLOS ONE. After careful consideration, we feel that it has merit but does not fully meet PLOS ONE’s publication criteria as it currently stands. Therefore, we invite you to submit a revised version of the manuscript that addresses the points raised during the review process.

The reviewers are satisfied with your revision, however the manuscript is still lacking a dedicated Limitations section, as requested in previous revisions. I would suggest the authors to use this section to discuss all limitations emerged during the review process. I would also invite the authors to work more to smooth their claims, especially about the Russian influence. 

We would appreciate receiving your revised manuscript by Jan 12 2020 11:59PM. To enhance the reproducibility of your results, we recommend that if applicable you deposit your laboratory protocols in protocols.io, where a protocol can be assigned its own identifier (DOI) such that it can be cited independently in the future. For instructions see: http://journals.plos.org/plosone/s/submission-guidelines#loc-laboratory-protocols

We look forward to receiving your revised manuscript.

Kind regards,

Fabiana Zollo, Ph.D.

Academic Editor

PLOS ONE

Reviewers' comments:

Reviewer's Responses to Questions

**Comments to the Author**

1. If the authors have adequately addressed your comments raised in a previous round of review and you feel that this manuscript is now acceptable for publication, you may indicate that here to bypass the “Comments to the Author” section, enter your conflict of interest statement in the “Confidential to Editor” section, and submit your "Accept" recommendation.

Reviewer #2: All comments have been addressed

Reviewer #3: All comments have been addressed

2. Is the manuscript technically sound, and do the data support the conclusions?

Reviewer #2: Partly

Reviewer #3: (No Response)

3. Has the statistical analysis been performed appropriately and rigorously? 

Reviewer #2: Yes

Reviewer #3: (No Response)

4. Have the authors made all data underlying the findings in their manuscript fully available?

Reviewer #2: Yes

Reviewer #3: (No Response)

5. Is the manuscript presented in an intelligible fashion and written in standard English?

Reviewer #2: Yes

Reviewer #3: (No Response)

6. Review Comments to the Author

Reviewer #2: The authors addressed my concerns. For the future I suggest to the authors to spend time in writing more exhaustive response to reviewers.

Reviewer #3: (No Response)

7. PLOS authors have the option to publish the peer review history of their article (what does this mean?). If published, this will include your full peer review and any attached files.

Reviewer #2: No

Reviewer #3: No

---

## [Editor Report · Decision Letter 3]

31 Dec 2019

Investigating Italian disinformation spreading on Twitter in the context of 2019 European elections

PONE-D-19-20362R3

Dear Dr. Pierri,

We are pleased to inform you that your manuscript has been judged scientifically suitable for publication and will be formally accepted for publication once it complies with all outstanding technical requirements.

With kind regards,

Fabiana Zollo, Ph.D.

Academic Editor

PLOS ONE
---

## [Editor Report · Acceptance letter]

3 Jan 2020

PONE-D-19-20362R3 

Investigating Italian disinformation spreading on Twitter in the context of 2019 European elections 

Dear Dr. Pierri:

I am pleased to inform you that your manuscript has been deemed suitable for publication in PLOS ONE. Congratulations! Your manuscript is now with our production department. 

With kind regards,

on behalf of

Dr. Fabiana Zollo 

Academic Editor

PLOS ONE